# Invasive and Non-Invasive Neuromodulation for the Treatment of Substance Use Disorders: A Review of Reviews

**DOI:** 10.3390/brainsci15070723

**Published:** 2025-07-06

**Authors:** Tyler S. Oesterle, Nicholas L. Bormann, Majd Al-Soleiti, Simon Kung, Balwinder Singh, Michele T. McGinnis, Sabrina Correa da Costa, Teresa Rummans, Mohit Chauhan, Juan M. Rojas Cabrera, Sara A. Vettleson-Trutza, Kristen M. Scheitler, Hojin Shin, Kendall H. Lee, Mark S. Gold

**Affiliations:** 1Department of Psychiatry and Psychology, Mayo Clinic, Rochester, MN 55905, USAbormann.nicholas@mayo.edu (N.L.B.); al-soleiti.majd@mayo.edu (M.A.-S.); singh.balwinder@mayo.edu (B.S.);; 2Mayo Clinic Libraries, Mayo Clinic, Rochester, MN 55905, USA; mcginnis.michele@mayo.edu; 3Department of Psychiatry and Psychology, Mayo Clinic, Jacksonville, FL 32224, USA; rummans.teresa@mayo.edu (T.R.);; 4Department of Neurologic Surgery, Mayo Clinic, Rochester, MN 55905, USAvettleson-trutza.sara@mayo.edu (S.A.V.-T.);; 5Department of Psychiatry, Washington University School of Medicine, Washington University in St. Louis, St. Louis, MO 63130, USA

**Keywords:** addiction medicine, substance-related disorders, neuromodulation, transcranial magnetic stimulation (rTMS), transcranial direct current stimulation (tDCS), and deep brain stimulation (DBS)

## Abstract

Background: Invasive and non-invasive neuromodulation in psychiatry represents a burgeoning field that leverages advanced neuromodulation techniques to address substance use disorders (SUDs). Aims: This narrative review synthesizes findings from multiple reviews to evaluate the efficacy of neuromodulation in treating SUDs. Methods: A comprehensive literature search was conducted between December 2024 and April 2025, focusing on systematic reviews and meta-analyses that examined various neuromodulation modalities, including repetitive transcranial magnetic stimulation (rTMS), transcranial direct current stimulation (tDCS), and deep brain stimulation (DBS). The selected reviews were analyzed to identify common themes, outcomes, and gaps in the current understanding of these treatments for SUDs. Results: 11 reviews met the final inclusion criteria; 5 focused on non-invasive neuromodulation (rTMS, tDCS) and 6 on invasive neuromodulation (DBS). Non-invasive neurostimulation was associated with modest improvements in craving and cognitive dysfunction in individuals with SUDs. Similarly, invasive neuromodulation (DBS), through high-frequency stimulation of the bilateral nucleus accumbens, appeared to reduce cravings and improve comorbid psychiatric symptoms in both preclinical and human studies. Importantly, small sample sizes, heterogeneity in targets and stimulation protocols, and short follow-up periods significantly limit the generalizability of current findings from both non-invasive and invasive neuromodulation studies. Conclusions: As novel and more effective therapies for the treatment of SUD are desperately needed, procedural interventional psychiatry holds promise. However, despite encouraging results, existing evidence is still preliminary, and larger, rigorously designed studies are warranted to further establish the safety and efficacy of neuromodulatory interventions for SUD treatment.

## 1. Introduction

Interventional procedural strategies to treat mental illness first emerged in the 1930s, when psychiatrists began to employ “somatotherapies” such as insulin-induced coma therapy, Metrazol (pentamethylenetetrazol) convulsive therapy, lobotomy, and electroconvulsive therapy (ECT) to treat severe psychiatric conditions. Among these, ECT emerged as the safest and most effective modality [1]. Over the following decades, advances in neuroscience and medical technology led to the development of more precise neuromodulation techniques, including deep brain stimulation (DBS), transcranial magnetic stimulation (TMS), and transcranial direct current stimulation (tDCS), which offer improved safety profiles and promise for individuals experiencing more severe psychiatric conditions, at times, refractory to conventional therapies and pharmacological interventions [2]. Concurrently, the conceptualization of addiction has evolved over the past decades. In the 1930s, addiction was often viewed as a “moral failing” [3] for which therapeutic interventions were limited. However, decades of research have demonstrated a robust neurobiological disease model encompassing multiple brain structures and related circuitry [4]. Despite significant advances in the understanding of substance use disorders (SUDs), these conditions remain highly disabling and are associated with significant morbidity and premature mortality. Existing interventions for SUD treatment remain suboptimal, and novel and more effective therapies are desperately needed.

Addiction is thought to occur in three stages: binge/intoxication, withdrawal/negative affect, and preoccupation/anticipation (or craving) [5]. These stages are mediated by discrete, reproducible neural circuits, primarily involving dopaminergic pathways, such as the mesolimbic (the ventral tegmental area to the nucleus accumbens), mesocortical (the ventral tegmental area to the prefrontal cortex), and mesostriatal (the substantia nigra to the dorsal striatum). Moreover, the orbitofrontal cortex and anterior cingulate cortex are involved in salience attribution and inhibitory control, while the amygdala and hippocampus contribute to memory formation and conditioned responses [6].

With these insights and the advent of novel technologies, interventional psychiatry has reemerged as a field that leverages procedural techniques targeting dysfunctional neural circuits for the treatment of psychiatric disorders [7,8,9]. Neuromodulation is a specific type of interventional psychiatry that involves the application of electrical or magnetic stimuli to influence connectivity and the functioning of a neural circuit or pathway. In contrast to modalities like ECT, the precision of neuromodulation strategies such as DBS and TMS offers the ability to target specific brain structures and systems. In the case of SUDs, both invasive and non-invasive neuromodulation therapies have targeted components of the meso-cortico-limbic pathways, including the ventral striatum (VS), nucleus accumbens (NAc), prefrontal cortex (PFC), anterior cingulate cortex (ACC), and others (Figure 1). In this review of reviews, we synthesize the current evidence on the efficacy of both invasive and non-invasive neuromodulation strategies for the treatment of SUDs.

To guide this synthesis, we structured our primary research question using the PICOT framework: In individuals with substance use disorders (Population), do neuromodulation techniques such as TMS, tDCS, or DBS (Intervention), compared to sham or no neuromodulation (Comparison), result in improved cravings, cognitive function, or relapse prevention (Outcomes) when assessed over the period during which participants were observed for outcomes in the included studies (Time)? Our primary objective was to characterize the effectiveness and limitations of each neuromodulation modality. A secondary objective was to identify consistent stimulation targets, common outcome domains, and methodological gaps across existing reviews.

## 2. Materials and Methods

We conducted a comprehensive search to find relevant literature on interventional approaches for substance use disorders. A medical librarian (M.T.M.) deployed the search strategy, developed for Embase, in consultation with the research team, and reviewed by an independent medical librarian. The Embase search was manually translated across all resources using syntax, controlled vocabulary, and search fields. MeSH thesaurus terms from MEDLINE, EMTREE thesaurus terms from Embase, Thesaurus of Psychological Index Terms, and free text words were used for search concepts. The full search strategy is available in the Appendix A.

Studies were identified from searches of the databases Embase, MEDLINE, Cochrane Central Register of Controlled Trials (CENTRAL), Cochrane Database of Systematic Reviews (CDSR), and PsycINFO- all via the Wolters Kluwer Ovid interface; Scopus; Science Citation Index Expanded (SCI-Expanded) and Emerging Sources Citation Index (ESCI)—both via the Clarivate Analytics Web of Science interface. English language database limits were applied as available or built into searches when possible. An unpublished filter was used to limit the study design to a systematic review with or without meta-analysis. Conference abstracts and conference reviews were excluded via a command-line search. Initial database searches were performed on December 20, 2024. Additional searches were deployed on 2 April 2025 to identify studies using electroconvulsive therapy and vagal nerve stimulation. The Appendix A provide full search strategies. All records were downloaded or manually added to EndNote 20 Desktop version and deduplicated using a method by Bramer et al. [10].

NLB and TSO screened all titles and abstracts for relevance and conducted full-text reviews for final inclusion. Disagreements were resolved by discussion, with MSG available as a third-party tie-breaker if needed. To minimize redundancy among included reviews, four additional reviewers (M.A., J.M.R.C., S.A.V.-T., K.M.S.) conducted an overlapping review analysis to identify and exclude earlier reviews that were fully contained within more recent or comprehensive ones. This team also extracted variables including the total number of studies and/or patients included, neuromodulation modality, stimulation parameters, target brain regions, substances studied, and reported outcomes such as craving, abstinence, cognitive effects, and relapse.

## 3. Results

We imported 542 references for screening. After removing 25 duplicates identified via Covidence, 517 studies remained for title and abstract screening. Of those, 450 were excluded based on irrelevance to the review criteria. The remaining 62 studies underwent full-text assessment for eligibility. Among full-text articles, we noted substantial overlap among non-invasive neuromodulation reviews. As a result, we excluded 31 older studies because they included studies that were already in more recent reviews. Additionally, 19 reviews were excluded for focusing on non-neuromodulation procedural interventions such as ketamine and various psychedelics. One review was excluded due to reporting outcomes not relevant to our review. In total, 51 reviews were excluded at the full-text stage. Finally, 11 reviews met the final inclusion criteria; 5 focused on non-invasive neuromodulation and 6 on invasive neuromodulation (Figure 2).

### 3.1. Non-Invasive Neuromodulation

Non-invasive brain stimulation techniques, such as tDCS and TMS, have gained attention for their potential to target specific brain regions associated with SUDs without the need for anesthesia. In addition to avoiding surgical risk and cost, these techniques offer greater accessibility, ease of use, and the potential for outpatient or even at-home administration, making them attractive for broader implementation in real-world addiction treatment settings. These techniques aim to modulate neural activity and improve cognitive functions, reduce cravings, and enhance treatment outcomes. TMS induces scalp and skull penetrating magnetic fields, modulating cortical neuron excitability. The depth and extent of stimulation depend on the coil type and stimulation parameters. Standard figure-8 coils can reach 1.5–2 cm beneath the skull, primarily affecting superficial cortical areas such as the dorsolateral prefrontal cortex (DLPFC). In contrast, H-coils used in deep TMS (dTMS) can stimulate midline and subcortical regions up to 4–5 cm deep, including the medial prefrontal cortex (mPFC), dorsal anterior cingulate cortex (dACC), and insula [11].

Stimulation frequency determines the direction of neuroplastic changes: high-frequency rTMS (≥5Hz, typically 10–20 Hz) generally increases cortical excitability, whereas low-frequency stimulation (≥1 Hz) and continuous theta burst stimulation tend to decrease excitability. Intermittent theta burst stimulation, a patterned high-frequency protocol, can mimic the effects of conventional high-frequency rTMS in shorter sessions [12].

Amerio et al. conducted a systematic review of eight randomized controlled trials evaluating the efficacy and safety of rTMS in treating cocaine use disorder. The strongest findings emerged from high-frequency (≥5 Hz) rTMS protocols targeting the left DLPFC, which significantly reduced self-reported cue-induced craving, impulsivity, and, in some cases, cocaine use compared to controls. Trials stimulating the mPFC or bilateral prefrontal cortices showed less consistent or no significant effects. Overall, rTMS appeared well-tolerated with no serious adverse events reported. However, the review highlighted several limitations: small sample sizes, heterogeneous stimulation protocols, high dropout rates, predominantly male samples limiting generalizability, exclusion of patients with psychiatric comorbidities, reliance on subjective craving measures, and challenges in achieving truly inert sham TMS conditions [13].

In the meta-analysis by Bormann et al., five randomized controlled trials combining TMS or tDCS with medication for addiction treatment for SUDs were analyzed. The pooled results showed a moderate, statistically significant reduction in craving-related measures (Hedges’ g = −0.42, 95% CI: −0.73 to −0.11, *p* = 0.008) compared to sham stimulation. However, the authors pooled across heterogeneous protocols, combining different stimulation modalities (rTMS and tDCS), stimulation frequencies (low, high, and theta burst), and both opioid and tobacco studies into a single analysis, limiting the ability to isolate effects specific to stimulation type, frequency, or substance. The authors also provided the rationale for combining medication-assisted treatment (MAT) with neuromodulation, arguing that their co-administration may work in a complementary way [14].

Zhang et al. conducted a systematic review evaluating the effectiveness of tDCS for improving cognitive dysfunction in patients with SUDs. Twenty-two sham-controlled studies with 770 participants were included, covering various substances (nicotine, alcohol, methamphetamine, cocaine, opioids, marijuana) and a range of cognitive outcomes (executive function, attention, risk-taking, impulsivity). High heterogeneity was observed across studies in terms of targeted brain regions, stimulation parameters (current intensity, density, duration), intervention designs (single vs. multiple sessions), and cognitive tasks used, precluding a meta-analysis. However, most studies targeted the DLPFC using standard or high-definition tDCS protocols. Overall, tDCS showed potential for improving executive function, attention, and reducing risk-taking behaviors, but results were inconsistent. Some studies reported significant cognitive benefits, while others showed no effect or even negative outcomes (e.g., increased risk-taking). Risk-of-bias assessments indicated generally good study quality, although many studies had unclear randomization and blinding procedures [15].

Sahaf et al. conducted a systematic review and meta-analysis to evaluate the effectiveness of tDCS in treating methamphetamine use disorder. In total, 14 studies were included in the systematic review, and 7 studies with 495 subjects were meta-analyzed. Of the 14 included studies, 7 stimulated the right DLPFC, 6 stimulated the left DLPFC, and 1 targeted the right cheek. A few studies targeted the inferior frontal gyrus (IFG) or the frontal-parietal-temporal (FPT) association area. Stimulation protocols varied considerably in session number (1–15 sessions), intensity (0.45–2 mA, most commonly 2 mA), and duration (10–30 min). tDCS demonstrated a moderate to large effect size in reducing cravings (Hedges’ g = −0.64, 95% CI: −0.85 to −0.30, *p* < 0.05) compared to sham stimulation. Improvements in working memory and cognitive flexibility were more often associated with left DLPFC stimulation, while inhibitory control outcomes were mixed across stimulation sites. Despite positive findings, the included studies had small sample sizes, variability in stimulation protocols, and inconsistent cognitive and craving measurement tools, limiting firm conclusions [16].

Chan et al. conducted a meta-analysis of 43 randomized sham-controlled trials (n = 2008 participants) evaluating tDCS for reducing cravings across various SUDs, including alcohol, opioids, methamphetamine, cocaine, tobacco, and cannabis. Overall, tDCS produced a moderate and statistically significant reduction in craving (SMD = −0.74, 95% CI: −0.98 to −0.50, *p* < 0.001), although substantial heterogeneity was observed across studies (I^2^ = 83%). Bilateral stimulation and anodal placement over the right dorsolateral prefrontal cortex (DLPFC), along with stimulation protocols using 1.5–2 mA current, 20 min sessions, and electrode sizes ≥ 35 cm^2^, were associated with stronger effects. Substance-specific analyses showed significant craving reductions for opioids, methamphetamine, cocaine, and tobacco, but not for alcohol or cannabis. However, the meta-analysis had several limitations: persistent high heterogeneity despite subgroup analyses, evidence of publication bias (especially in opioid and tobacco studies), small sample sizes, variability in craving assessment tools, and inconsistent control for mood-related confounders [17].

In summary, non-invasive neuromodulation, particularly tDCS and rTMS targeting the DLPFC, was associated with modest reductions in craving and, in some cases, improvements in cognitive function. Craving-related improvements were most consistently reported in studies targeting opioid, stimulant, and tobacco use disorders, particularly in reviews employing meta-analytic methods. However, findings for alcohol and cannabis were less consistent, and results for cognitive outcomes such as executive function, attention, and impulsivity were mixed. Across reviews, the DLPFC was the most frequently targeted region, with both left- and right-sided stimulation protocols explored. Several reviews suggested greater benefit with high-frequency rTMS and bilateral or right-sided tDCS, but stimulation parameters, electrode configurations, and session protocols varied widely. Overall, limitations such as small sample sizes, protocol heterogeneity, subjective outcome measures, and short follow-up periods were common and significantly limit the generalizability of current findings (see Table 1 for results summary).

### 3.2. Invasive Neuromodulation for SUD

DBS involves the surgical implantation of electrodes into specific brain regions to deliver targeted electrical stimulation to modulate brain function. The current is transmitted from an infraclavicular implantable pulse generator via subcutaneous extension wires to the brain. Stimulation parameters such as amplitude, frequency, and pulse width can be adjusted based on the therapeutic target and the intended modulation of neural activity within the implicated circuit [18]. Because of its success and excellent safety profile in treating movement disorders for three decades, DBS has been investigated as a potential intervention for other disorders involving dysfunctional neurocircuitry, such as SUDs. Its application in addiction treatment was promoted by preclinical evidence demonstrating reduced drug-seeking behaviors following NAc disruption, as well as serendipitous clinical observations of reduced substance use in patients receiving DBS to the NAc for other indications [19].

A systematic review by dos Santos et al. (2020) analyzed 12 studies involving 21 patients with primarily heroin, alcohol, or cocaine use. Following high-frequency stimulation (130–185 Hz) to the bilateral NAc (with or without stimulation of the anterior limb of the internal capsule, ALIC), 52% of participants experienced relapse, though often with reduced frequency or severity compared to pre-DBS [20]. Hassan et al. (2020) reviewed 14 studies involving 33 patients treated with high-frequency bilateral NAc +/− ALIC DBS for heroin, alcohol, nicotine, cocaine, or methamphetamine use. Remission rates were 61% at six months and 53% at one year. Most studies were case reports, with follow-up durations ranging from ten months to eight years [21]. Navarro et al. (2022) compared eight studies, including thirteen patients using high-frequency bilateral DBS at the NAc, finding similar relapse rates among SUDs (38.4% overall; 25% opioid, 50% alcohol, and 33.3% nicotine) [22].

Fattahi et al. (2020) systematically reviewed both preclinical and clinical studies of bilateral high-frequency DBS to the NAc for opioid use disorder, including nine human studies (16 patients) and 12 animal studies. The authors reported overall decreased craving and consumption in both humans and animals [23]. Shaheen et al. (2023) conducted a meta-analysis of 16 studies (50 patients) of bilateral high-frequency DBS to the NAc +/− ALIC for various SUD. They reported a mean “effect size” of 55.9 (95% CI: 40.4–71.4) for reductions in craving and consumption, though this value likely reflects a percentage reduction or raw score difference, not a standardized Cohen’s d effect size as the study reports. Despite this limitation in reporting, the study estimated a mean clinical improvement of 59.6% and a relapse rate of just 8%, the lowest among all reviews. Also unique to this study was their subgroup analysis, which highlighted differences in outcomes based on addiction duration (a longer duration indicated a higher likelihood of relapse), patient age (>45 years responded better to DBS), and type of drug (DBS was more effective for alcohol and opioid use than for nicotine). This was the first meta-analysis on the topic and suggested that DBS is effective for SUD treatment, but there was substantial heterogeneity among studies [24]. Dimech et al. (2024) systematically reviewed 26 studies, including the first randomized controlled trial of DBS for SUD. While targeting the bilateral NAc +/− ALIC reduced cravings across all substances, the relapse rate was 73.2%—the highest reported among reviews [25].

All reviews investigated the high-frequency stimulation of the bilateral NAc and generally reported reductions in cravings and comorbid psychiatric symptoms in both preclinical and human studies. However, significant heterogeneity was noted across studies, with outcome variations based on methodology and substance type. DBS appeared most effective for alcohol and opioid use disorders, with relapse rates similar to those seen with NAc lesioning. Several limitations restricted comparability across reviews: (1) despite all studies targeting the NAc, coordinates were inconsistently reported, with variation among anatomic reference points; (2) follow-up times were highly variable, ranging from three months to eight years; (3) the risk of publication bias was inconsistently assessed among studies (e.g., a lack of standardization across studies in how cravings were measured); (4) no standardized outcome or definition was reported across studies; (5) the potential causes of relapse among subjects were inconsistently assessed or documented (e.g., relapses due to external stressors vs. relapses due to desire or craving for the drug); and (6) studies were primarily case reports, limited by small numbers, and lacked blinded stimulation. A summary of these reviews is provided in Table 2.

## 4. Discussion

This review of reviews synthesizes the current evidence on both invasive and non-invasive neuromodulation techniques for the treatment of SUDs. Across the 11 included reviews, several consistent patterns emerged. The non-invasive neuromodulation techniques tDCS and rTMS were most commonly applied to the DLPFC, while invasive approaches uniformly targeted the NAc. Craving was the most frequently assessed outcome and showed the clearest evidence of improvement across both modalities, with effects particularly evident in studies targeting opioid, stimulant, and tobacco use disorders. In contrast, findings related to alcohol and cannabis were more variable. High-frequency rTMS and bilateral or right-sided tDCS appeared more effective than low-frequency or unilateral approaches, and stimulation protocols using 1.5–2 mA over 20 min sessions showed stronger effects in meta-analyses [13,17]. Although some reviews also assessed cognitive outcomes, including attention, executive function, and risk-taking, these findings were mixed and less robust than craving-related effects [15,16]. Together, the results suggest that neuromodulation holds promise for the targeting of the neural circuits underlying addiction-related behaviors, with emerging evidence for both symptom improvement and circuit-specific modulation.

These synthesized findings are largely consistent with the broader neuromodulation literature in psychiatry and build upon prior umbrella reviews and meta-analyses by focusing specifically on SUD. Several included reviews align closely with earlier work in depression, cognitive neuroscience, and circuit-based psychiatric models. For example, Naish et al. concluded that while tDCS and rTMS may improve executive function in individuals with addiction, generalizability was limited by the exclusion of participants with psychiatric comorbidities in many primary studies [26]. These findings parallel the methodological concerns noted in reviews by Zhang et al. and Sahaf et al., which examined cognitive domains including working memory, attention, impulsivity, and risk-taking. Both reviews reported heterogeneous effects: some studies demonstrated cognitive benefits—particularly for executive function and attention—while others showed null or inconsistent findings [15,16]. Zhang et al. commented on seemingly paradoxical outcomes, such as increased risk-taking, underscoring the variability in tDCS effects across cognitive domains [15]. Sahaf et al. found that working memory and cognitive flexibility tended to improve, especially with left DLPFC stimulation, while inhibitory control outcomes were more mixed [16].

Further insights into stimulation protocols come from Razza et al. who conducted an umbrella review of neuromodulation techniques for major depressive disorder. They found that high-frequency rTMS targeting the left DLPFC and bilateral tDCS had the strongest supporting evidence, with consistent improvements in clinical outcomes [27]. A similar pattern of left DLPFC targeting has emerged in the addiction literature, including the review by Amerio et al. in individuals with cocaine use disorder [13]. In contrast, a meta-analysis by Chan et al. examining tDCS for craving reduction in SUDs highlighted slightly different parameters, reporting stronger effects with bilateral stimulation and anodal placement over the right DLPFC [17]. These laterality differences may reflect distinctions in symptom targets (mood vs. craving), stimulation modality (rTMS vs. tDCS), or substance-specific neurocircuitry. Brunoni et al., in an individual patient data meta-analysis of tDCS for depression, demonstrated that higher stimulation “doses” (defined by current intensity and the number of sessions) were associated with greater treatment response [28]. Although conducted in the context of major depressive disorder, their findings reinforce the broader principle that stimulation parameters significantly influence neuromodulation efficacy. Together, this underscores that regardless of condition or modality, the clinical response depends heavily on stimulation configuration and delivery. Similar principles apply to invasive neuromodulation: reviews of DBS for SUDs consistently targeted the NAc and related cortico-striatal circuits, mirroring approaches in mood and compulsivity disorders [29,30]. These findings support the broader application of circuit-based models in addiction psychiatry, though current evidence remains limited by small samples and protocol heterogeneity.

Although neuromodulation is not broadly included in clinical guidelines for SUD treatment, rTMS has received FDA approval for smoking cessation, marking an important regulatory milestone. However, its clinical adoption remains limited partly due to inconsistent insurance coverage, the lack of standardized reimbursement models, and the overall investment required (both financial and logistical) to offer the intervention. As the evidence base grows, broader integration into treatment guidelines will depend on the accumulation of large, well-controlled trials that clarify optimal protocols, long-term effectiveness, and real-world feasibility.

Several limitations should be considered when interpreting these findings. First, as a review of reviews, our analysis is inherently constrained by the scope, quality, and overlap of the included reviews. While we aimed to reduce redundancy by excluding overlapping publications, some overlap of primary studies did occur, which may have led to the implicit duplication of findings. Second, we did not conduct a formal risk of bias assessment, as our primary aim was to provide a narrative synthesis of outcomes and methodological patterns rather than evaluate the internal validity of each review. Nonetheless, we reported common limitations described within the included reviews, such as small sample sizes, short follow-up periods, protocol heterogeneity, and publication bias. Third, the heterogeneity in stimulation targets, protocols, outcome measures, and substance types across reviews limits the generalizability of conclusions. Finally, most DBS studies were small, open-label case series conducted at a single center, further limiting confidence in those findings.

Despite these constraints, this review highlights important directions for future research. The standardization of stimulation protocols, clearer reporting of cognitive and functional outcomes, and inclusion of diverse participant samples—including those with psychiatric comorbidities—are needed to improve generalizability. Large, multi-site trials of both invasive and non-invasive neuromodulation approaches will be critical to establish replicable effects across substance types and treatment-refractory populations [2]. Additionally, future studies should incorporate rigorous methods for assessing the durability of effects, cost-effectiveness, and real-world feasibility to inform clinical implementation. Outcomes such as craving have been shown to change in parallel with improvements in psychosocial domains [31]. As neuromodulation may also influence affective symptoms and stress reactivity [9,27], future research should examine how these psychosocial dynamics interact with neural circuit modulation to influence treatment outcomes. While neuromodulation has been increasingly studied as a maintenance treatment for SUDs, its application during acute withdrawal remains understudied [32,33]. Given emerging evidence of benefit during maintenance treatment, further exploration of these approaches in the withdrawal context may be warranted.

## 5. Conclusions

Across 11 systematic reviews, it was found that the modalities TMS, tDCS, and DBS were associated with modest reductions in craving, and in some cases, cognitive improvements. These effects were most consistently demonstrated in studies involving opioid, stimulant, and tobacco use disorders, while findings for alcohol and cannabis were less consistent. Reported stimulation targets frequently included the DLPFC, NAc, and ACC(See Figure 2 for a visual depiction of target locations). By synthesizing findings across both invasive and non-invasive approaches, this review provides a broad, integrative perspective on the potential for neuromodulation to target dysfunctional neural circuits involved in addiction. It also highlights key methodological gaps that limit the current evidence base, including small sample sizes, heterogeneous stimulation protocols, short follow-up periods, and a lack of sham-controlled studies, particularly in the DBS literature. These findings can inform future trial design, refine candidate targets, and support the ethical and effective integration of neuromodulation into treatment frameworks for SUDs as the field advances. Future research should focus on standardizing protocols, optimizing stimulation parameters, and integrating neuromodulation with behavioral and pharmacological interventions to maximize efficacy. Despite encouraging results, the field requires larger, rigorously designed sham-controlled studies to establish clinical effectiveness. Importantly, these interventions’ ethical implications and safety must remain central considerations as their clinical use expands.

## Figures and Tables

**Figure 1 brainsci-15-00723-f001:**
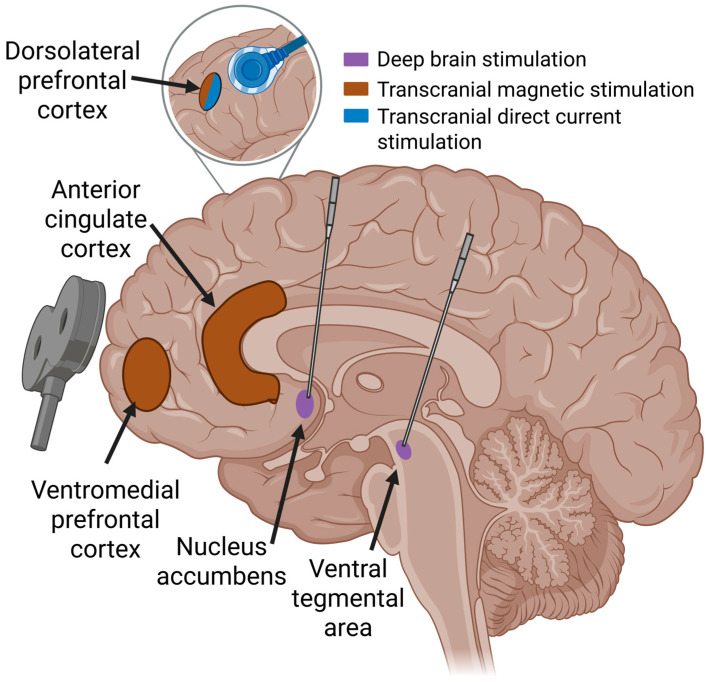
Neuromodulation targets and methods. This schematic illustrates key brain regions targeted by neuromodulation for substance use disorders. Transcranial magnetic stimulation (brown) is applied non-invasively to brain regions, including the dorsolateral prefrontal cortex, using magnetic coils, such as the commonly used figure-8 coil. Transcranial direct current stimulation (blue) also frequently targets the dorsolateral prefrontal cortex and is delivered via electrodes placed on the scalp. Deep brain stimulation (purple) involves the surgical implantation of electrodes into subcortical regions such as the nucleus accumbens or ventral tegmental area. The figure highlights the variation in delivery methods and target depth across neuromodulation approaches.

**Figure 2 brainsci-15-00723-f002:**
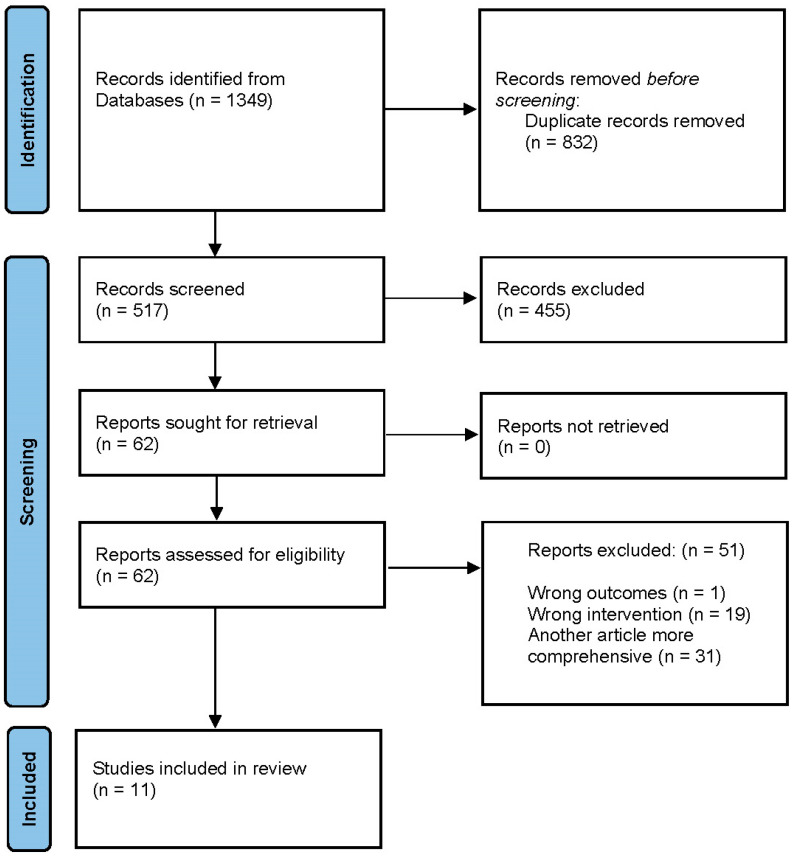
PRISMA flow diagram of included reviews. This diagram outlines the screening and selection process for included reviews.

**Table 1 brainsci-15-00723-t001:** Summary of systematic reviews or meta-analyses regarding non-invasive brain stimulation for substance use disorders.

Authors	Type of Study	Type of Neuromodulation	Target Location	Target Substance of Abuse	Findings
Amerio et al. [13]	Systematic review	rTMS	Left DLPFC	Cocaine	High-frequency rTMS significantly reduced self-reported cue-induced cocaine craving and impulsivity
Chan et al. [17]	Meta-analysis	tDCS	Right DLPFC	Alcohol, opioids, methamphetamine, cocaine, tobacco, cannabis	tDCS led to a moderate reduction in cravings for opioids, methamphetamine, cocaine, and tobacco
Sahaf et al. [16]	Systematic review and meta-analysis	tDCS	DLPFC	Methamphetamine	tDCS significantly reduced craving
Zhang et al. [15]	Systematic review	tDCS	DLPFC	Alcohol, nicotine, cocaine, methamphetamine, opioids, cannabis	tDCS can improve cognitive functions in SUD patients
Bormann et al. [14]	Systematic review and meta-analysis	TMS or tDCS	DLPFC	Opioids, tobacco	Combining TMS or tDCS with MAT significantly reduced craving-related measures compared to sham stimulation, with a Hedges’ g effect size of −0.42 (95% confidence interval: −0.73 to −0.11, *p* < 0.01).

Abbreviations: DLPFC: Dorsolateral prefrontal cortex; MAT: Medication-assisted treatment; rTMS: Repetitive transcranial magnetic stimulation; SUD: Substance use disorder; tDCS: Transcranial direct current stimulation; TMS: Transcranial magnetic stimulation.

**Table 2 brainsci-15-00723-t002:** Summary of systematic reviews or meta-analyses regarding deep brain stimulation for substance use disorders.

Author	Substance Use Disorder	Inclusion/Exclusion Criteria	Intervention	Results
dos Santos (2020) [20]	Various—heroin (62%), alcohol (33%), cocaine (5%), others	Inclusion: DBS in patients with SUDs, with no restriction on gender, age, ethnicity, or substance Exclusion: DBS without SUD as main focus; discontinued clinical trials	High-frequency DBS to NAc ± additional stim to ALIC, VS/VS, BNST	Reduced craving in most patients. In total, 48% achieved abstinence; relapses typically occurred with reduced severity. Some mood and quality of life improvements.
Hassan (2020) [21]	Various—heroin (43%), alcohol (29%), nicotine (14%), cocaine, meth	Inclusion: Human studies targeting NAc Exclusion: follow-up less than 6 mos.	High-frequency DBS to NAc ± ALIC	Reduced cravings. Remission: 61% at 6 months, 53% at 1 year, 43% at 2 years; relapse common over time.
Navarro (2020) [22]	Various –Alcohol, opioids, nicotine	Inclusion: NAc DBS or lesioning Exclusion: articles with dual NAc/ALIC DBS	High-frequency DBS to NAc vs. lesioning via RF ablation	Reduced cravings. Relapse: 38.4% with DBS and 39% for RF ablation; further studies needed
Fattahi (2022) [23]	Opioids	Inclusion: Studies on opioid-dependent individuals, both human and animal models	High-frequency DBS targeting NAc ± ALIC, VC/VS	Reduced opioid craving and consumption. Abstinence not consistently reported. Supports DBS as a promising alternative
Shaheen (2023) [24]	Various—Alcohol, heroin, tobacco	Inclusion: DBS for SUD Exclusion: did not include any scale of addiction treatment	HF DBS to NAc ± ALIC	Reduced cravings—59.6% clinical improvement; relapse rate 8%. Outcomes vary by age, substance type, and addiction duration
Dimech (2024) [25]	Various	Inclusion: DBS for SUD Exclusions: non-SUD	HF DBS to NAc ± ALIC	Reduced cravings. Relapse of 73.2%. Psychiatric symptom improvements noted.

Across studies, stimulation parameters most commonly used were 130–185 Hz frequency, 90–240 µs pulse width, and 1–7 V amplitude. Outcome measures varied across reviews, with craving reduction, abstinence, relapse, or clinical improvement reported depending on each review’s focus. Quantitative estimates for craving were inconsistently reported and often embedded within broader clinical outcomes. Abbreviations: ALIC: Anterior limb of the internal capsule; BNST: Bed nucleus of the stria terminalis; DBS: Deep brain stimulation; HF: High frequency; NAc: Nucleus accumbens; RF: Radio frequency; SUD: Substance use disorder; VC/VS: Ventral capsule/ventral striatum; VS/VS: Ventral striatum/ventral striatum.

## Data Availability

The data presented in this study are derived from previously published articles and are publicly available in the respective publications cited throughout the manuscript.

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
