# Peer review of "Invasive and Non-Invasive Neuromodulation for the Treatment of Substance Use Disorders: A Review of Reviews"

_brainsci, 2025, doi:10.3390/brainsci15070723_

Round 1

Reviewer 1 Report

Comments and Suggestions for Authors

The authors submit a comprehensive review of reviews for invasive and non-invasive neuromodulation for treatment of substance use disorder. The methodology and scope of conclusions are appropriate. There are a few things which could be improved upon before publication:

  1. Lines 65-67. The grammar is confusing, it is unclear what is trying to be said. 
  2. Figures 1 and 2 could benefit from more comprehensive descriptions in the figure text (rather than only in the manuscript body).
  3. Line 117-118. "...without requiring the added time and cost associated with anesthesia." There are several reasons that a non-invasive neuromodulatory technique may be beneficial. The authors argument would be stronger if these were stated rather than just cost associated with anesthesia.
  4. Lines 171-173. I'm not sure if a draft from a comment somehow made it into the text? These lines should be clarified. 
  5. Line 213. "...desired neuromodulation." is vague.
  6. Table 2. The results table is a bit inconsistent, some citing remission rates, relapse rate, no rates, etc. I'm wondering if the findings could be made more consistent?
  7. Figure 2 is very nice. A lot of the text discusses tDCS in parallel with TMS. . If this is viewed as equally relevant to the authors for non-invasive neuromodulation as TMS, is it possible to incorporate tDCS into the figure?

Author Response

Reviewer 1

The authors submit a comprehensive review of reviews for invasive and non-invasive neuromodulation for treatment of substance use disorder. The methodology and scope of conclusions are appropriate. There are a few things which could be improved upon before publication:

  1. Lines 65-67. The grammar is confusing, it is unclear what is trying to be said. 

>> RESPONSE: Thank you for highlighting this, we agree. We have edited this to the following: “Moreover, the orbitofrontal cortex and anterior cingulate cortex are involved in salience attribution and inhibitory control, while the amygdala and hippocampus contribute to memory formation and conditioned responses.”

  1. Figures 1 and 2 could benefit from more comprehensive descriptions in the figure text (rather than only in the manuscript body).

>> RESPONSE: Based on other reviewer feedback, the order of the figures is now flipped. We have also updated their titles. Revision Figure 2 (originally Figure 1) is now titled “PRISMA flow diagram of included reviews.” With further description of “This diagram outlines the screening and selection process for included reviews.” For revision Figure 1 (originally Figure 2), it is titled “Neuromodulation targets and methods” with a description of “This schematic illustrates key brain regions targeted by neuromodulation for substance use disorders. Transcranial magnetic stimulation (brown) is applied non-invasively to brain regions, including the dorsolateral prefrontal cortex, using magnetic coils, such as the commonly used figure-8 coil. Transcranial direct current stimulation (blue) also frequently targets the dorsolateral prefrontal cortex and is delivered via electrodes placed on the scalp. Deep brain stimulation (purple) involves the surgical implantation of electrodes into subcortical regions such as the nucleus accumbens or ventral tegmental area. The figure highlights the variation in delivery methods and target depth across neuromodulation approaches.

  1. Line 117-118. "...without requiring the added time and cost associated with anesthesia." There are several reasons that a non-invasive neuromodulatory technique may be beneficial. The author's argument would be stronger if these were stated rather than just the cost associated with anesthesia.

>> RESPONSE: We agree with your critique. We have edited that sentence slightly and added with following immediately after it: “In addition to avoiding surgical risk and cost, these techniques offer greater accessibility, ease of use, and the potential for outpatient or even at-home administration, making them attractive for broader implementation in real-world addiction treatment settings.”

  1. Lines 171-173. I'm not sure if a draft from a comment somehow made it into the text? These lines should be clarified. 

>> RESPONSE: Thank you for this catch, and we apologize for that error. We have now deleted the part that said “21 of 22 (why 23 above?)” and double-checked and corrected everything to be consistent with the original literature and also the text throughout.

  1. Line 213. "...desired neuromodulation." is vague.

>> RESPONSE: We agree that the phrasing was vague and a bit circular. We have now replaced “desired neuromodulation” with “the intended modulation of neural activity within the implicated circuit.”

  1. Table 2. The results table is a bit inconsistent, some citing remission rates, relapse rate, no rates, etc. I'm wondering if the findings could be made more consistent?

>> RESPONSE: We have worked to make the results column more consistent, as we do agree with the reviewer. However, it is challenging due to heterogeneity across reviews. To aid in this, we also added to the table footnote: “Outcome measures varied across reviews, with craving reduction, abstinence, relapse, or clinical improvement reported depending on each review’s focus. Quantitative estimates for craving were inconsistently reported and often embedded within broader clinical outcomes.”.

  1. Figure 2 is very nice. A lot of the text discusses tDCS in parallel with TMS. . If this is viewed as equally relevant to the authors for non-invasive neuromodulation as TMS, is it possible to incorporate tDCS into the figure?

>> RESPONSE: Thank you for this suggestion; we have added tDCS to the image as suggested.

Reviewer 2 Report

Comments and Suggestions for Authors

Dear Editor, 

Please find below my review of the manuscript draft titled "Invasive and Non-Invasive Neuromodulation for the Treatment of Substance Use Disorders: A Review of Reviews", which has been submitted to the Brain Sciences journal.

The review evaluates the effectiveness of neuromodulation techniques in treating substance use disorders by synthesizingfindings from 11 previous reviews of both invasive and non-invasive methods. The results showed that non-invasive approaches, like rTMS and tDCS, offer modest improvements in craving and cognitive function. The most common potential brain targets included the dorsolateral prefrontal cortex, anterior cingulate cortex, nucleus accumbens, and insula. The review concludes that neuromodulation may serve as a beneficial complementary treatment for SUDs, though further research is necessary to fully elucidate its potential.

It is an interesting review on a very important and complex topic. The description of the existing background knowledge is adequate, while the literature search is exceptional. Overall, the manuscript draft is well-written and supported by the appropriate references, figures, and tables.

However, several concerns need to be addressed. A "review-of-reviews" is much more than just a compilation of individual review summaries.

  1. The search questions are ill-defined. The authors could use the PICOT frame to structure their primary and secondary research question(s) in an evidence-based manner.
  2. The results could be rationally organised according to primary and secondary research question(s).  
  3. The study also lacks a critical approach to the gathered literature. This limitation could be addressed using established tools, such as RoB-2, ROBIS, and AMSTAR for each review. 
  4. The discussion lacks depth and requires significant revision.
  5. A limitations section that outlines all constraints is necessary.
  6. In the conclusions, the authors must clearly state what this study adds to the existing literature to justify the publication of the current work.

The anticipated changes are going to alter the manuscript draft significantly. Therefore, I recommend a major revision.

Best regards

Author Response

Reviewer 2

The review evaluates the effectiveness of neuromodulation techniques in treating substance use disorders by synthesizing findings from 11 previous reviews of both invasive and non-invasive methods. The results showed that non-invasive approaches, like rTMS and tDCS, offer modest improvements in craving and cognitive function. The most common potential brain targets included the dorsolateral prefrontal cortex, anterior cingulate cortex, nucleus accumbens, and insula. The review concludes that neuromodulation may serve as a beneficial complementary treatment for SUDs, though further research is necessary to fully elucidate its potential.

It is an interesting review on a very important and complex topic. The description of the existing background knowledge is adequate, while the literature search is exceptional. Overall, the manuscript draft is well-written and supported by the appropriate references, figures, and tables.

However, several concerns need to be addressed. A "review-of-reviews" is much more than just a compilation of individual review summaries.

  1. The search questions are ill-defined. The authors could use the PICOT frame to structure their primary and secondary research question(s) in an evidence-based manner.

>> RESPONSE: Thank you for this suggestion; we have added the following to the end of the introduction: “To guide this synthesis, we structured our primary research question using the PICOT framework: In individuals with substance use disorders (Population), do neuromodulation techniques such as TMS, tDCS, or DBS (Intervention), compared to sham or no neuromodulation (Comparison), result in improved cravings, cognitive function, or relapse prevention (Outcomes) when assessed over the period during which participants were observed for outcomes in the included studies (Time)? Our primary objective was to characterize the effectiveness and limitations of each neuromodulation modality. A secondary objective was to identify consistent stimulation targets, common outcome domains, and methodological gaps across existing reviews.”

  1. The results could be rationally organized according to primary and secondary research question(s).  

>> RESPONSE: We appreciate the reviewer’s suggestion on ways to better align the Results section with the study’s primary and secondary research questions. To maintain a clear and accessible structure, we opted to organize the Results by intervention modality (non-invasive vs. invasive neuromodulation) and presented the findings review-by-review, ensuring that both primary (e.g., craving, cognitive function, relapse) and secondary (e.g., stimulation targets, protocols) elements were captured together for each included review. This approach allowed us to avoid presenting the same review multiple times under different outcome headings, thereby keeping the manuscript more concise, streamlined, and likely more accessible to the reader. In an attempt to address the reviewer’s comment directly, we revised the final paragraph of Results subsection 3.1 to explicitly synthesize consistent patterns, contradictions, and methodological limitations across reviews, structured in alignment with our primary and secondary aims.

However, in the section on invasive neuromodulation (DBS), all included studies targeted the bilateral nucleus accumbens. Given this consistency, we felt that subdividing results by outcome (e.g., craving vs. relapse) would have introduced artificial segmentation without improving clarity. We believe our revised structure offers a focused and coherent summary of findings, while still addressing the intent of the reviewer’s comment.

  1. The study also lacks a critical approach to the gathered literature. This limitation could be addressed using established tools, such as RoB-2, ROBIS, and AMSTAR for each review. 

>> RESPONSE: As this work was a scoping review-of-reviews rather than a systematic review or meta-analysis of primary studies, our objective was to provide a high-level synthesis of the current evidence base across both invasive and non-invasive neuromodulation modalities, rather than to formally rate the methodological rigor of each included review using tools such as RoB-2, ROBIS, or AMSTAR. However, we have tried to address this concern in two ways. First, we have updated the final paragraphs of the Results subsection, Sections 3.1, to more clearly highlight methodological limitations reported within the included reviews, such as small sample sizes, heterogeneity in protocols, short follow-up periods, and publication bias. Second, we have expanded the Limitations section of the Discussion to acknowledge the absence of a formal risk of bias assessment and to justify our approach in the context of the study’s aims and scope.

  1. The discussion lacks depth and requires significant revision.

>> RESPONSE: We revised the Discussion to address this concern by expanding our synthesis of findings across reviews, clarifying consistent patterns (e.g., craving-related outcomes were more robust than cognitive outcomes), and acknowledging modality- and symptom-specific variability (e.g., rTMS vs. tDCS; mood vs. craving). We incorporated external context from relevant umbrella reviews and meta-analyses in depression (e.g., Razza, Brunoni) and SUDs (e.g., Amerio, Chan) to situate our findings in the broader neuromodulation literature. Additionally, we added a discussion of existing FDA approval of rTMS for smoking cessation to improve clinical relevance and interpretive depth. This section is now significantly lengthened.

  1. A limitations section that outlines all constraints is necessary.

>> RESPONSE: We added a dedicated limitations paragraph to the Discussion. This section outlines key methodological constraints, including the inherent overlap across some reviews, the absence of formal risk of bias assessment, heterogeneity in stimulation parameters and outcome measures, and the preliminary nature of the DBS evidence base. We also used this section to identify future research priorities such as standardization of protocols, inclusion of more diverse participant populations, and the need for larger, multisite trials to improve generalizability and implementation feasibility.

  1. In the conclusions, the authors must clearly state what this study adds to the existing literature to justify the publication of the current work.

>> RESPONSE: We have re-worked the conclusion and added specific areas where this paper fills a gap. The bulk of this was accomplished by adding “By synthesizing findings across both invasive and non-invasive approaches, this review provides a broad, integrative perspective on the potential for neuromodulation to target dysfunctional neural circuits involved in addiction. It also highlights key methodological gaps that limit the current evidence base, including small sample sizes, heterogeneous stimulation protocols, short follow-up periods, and a lack of sham-controlled studies, particularly in the DBS literature. These findings can inform future trial design, refine candidate targets, and support the ethical and effective integration of neuromodulation into treatment frameworks for SUDs as the field advances.” into the existing conclusion.

Reviewer 3 Report

Comments and Suggestions for Authors

Formatting Revisions Needed:

1-Title formatting: The word “title” should be removed.

2- Line 21: Missing space → “SUDs. Methods:” should be “SUDs. Methods:”

3- General recommendation: Please check manuscript for consistent:

-Abbreviations (define at first use)

-Heading capitalization

-Punctuation spacing

For detailed feedback and specific recommendations, please refer to the attached document containing comprehensive comments and suggestions for improvement.

Author Response

Reviewer 3

Formatting Revisions Needed:

1-Title formatting: The word “title” should be removed.

>> RESPONSE: This has been corrected.

2- Line 21: Missing space → “SUDs. Methods:” should be “SUDs. Methods:”

>> RESPONSE: This has been corrected.

3- General recommendation: Please check manuscript for consistent:

-Abbreviations (define at first use)

>> RESPONSE: we have worked to correct this, except on occasion where it is displayed within a parenthesis. Abbreviating here will lead to a double parathesis and may be confusing for the reader. Therefore, we are happy to comply with whatever the publisher's preference is.

-Heading capitalization

>> RESPONSE: this has been made consistent

-Punctuation spacing

>> RESPONSE: this has been made consistent

For detailed feedback and specific recommendations, please refer to the attached document containing comprehensive comments and suggestions for improvement.

Introduction

  • Line 31: Introduce this acronym (NAc)

>> RESPONSE: This has been corrected.

  • Line 78: Authors are recommended to add a visual demonstration summarizing the available neuromodulation strategies for the treatment of SUDs.

>> RESPONSE: We have added tDCS based on another comment to Figure 2. Therefore, to address this specific comment, we have now moved Figure 2 earlier in the paper. Now, the updated Figure 2 (with the brain and the methods of DBS, rTMS, tDSC) is the revised Figure 1; and the PRISMA diagram is now the revised Figure 2.

Methods

  • PICOS Framework: Please clearly specify the Population, Intervention/Exposure, Comparison, Outcomes, and Study Design (PICOS) criteria used to guide the review.

>> RESPONSE: Thank you for this suggestion; we have added the following to the end of the introduction: “To guide this synthesis, we structured our primary research question using the PICOT framework: In individuals with substance use disorders (Population), do neuromodulation techniques such as TMS, tDCS, or DBS (Intervention), compared to sham or no neuromodulation (Comparison), result in improved cravings, cognitive function, or relapse prevention (Outcomes) when assessed over the period during which participants were observed for outcomes in the included studies (Time)? Our primary objective was to characterize the effectiveness and limitations of each neuromodulation modality. A secondary objective was to identify consistent stimulation targets, common outcome domains, and methodological gaps across existing reviews.”

  • Search Strategy: The keywords/search terms should be provided (possibly as supplementary material) to ensure transparency.

>> RESPONSE: these are now uploaded as supplementary materials; thank you for noting their absence.

  • An end date for the literature search should be included.
  • >> RESPONSE: within the methods, it now flows like this: “Initial database searches were performed on December 20, 2024. Additional searches were deployed on April 02, 2025 to identify studies using electroconvulsive therapy and vagal nerve stimulation.”
  • Quality Threshold for Included Reviews: Please state whether a minimum quality threshold (e.g., AMSTAR-2 score) was applied and how low-quality reviews were handled (e.g., excluded or sensitivity analysis).

>> RESPONSE: As this work was a scoping review-of-reviews rather than a systematic review or meta-analysis of primary studies, our objective was to provide a high-level synthesis of the current evidence base across both invasive and non-invasive neuromodulation modalities, rather than to formally rate the methodological rigor of each included review using tools such as RoB-2, ROBIS, or AMSTAR. However, we have tried to address this concern in two ways. First, we have updated the final paragraphs of the Results subsection, Section 3.1, to more clearly highlight methodological limitations reported within the included reviews, such as small sample sizes, heterogeneity in protocols, short follow-up periods, and publication bias. Second, we have expanded the Limitations section of the Discussion to acknowledge the absence of a formal risk of bias assessment and to justify our approach in the context of the study’s aims and scope.

  • Selection & Data Extraction Process: Details on screening (e.g., number of reviewers, resolution of disagreements) and data extraction (e.g., predefined variables, pilot testing of forms) are currently missing.
  • >> RESPONSE: We apologize for this oversight. We have added these specifics to the Materials and Methods section: “NLB and TSO screened all titles and abstracts for relevance and conducted full-text reviews for final inclusion. Disagreements were resolved by discussion, with MSG available as a third-party tie-breaker if needed. To minimize redundancy among included reviews, four additional reviewers (MNA, JMC, SAV, KMS) conducted an overlapping review analysis to identify and exclude earlier reviews that were fully contained within more recent or comprehensive ones. This team also extracted variables including the total number of studies and/or patients included, neuromodulation modality, stimulation parameters, target brain regions, substances studied, and reported outcomes such as craving, abstinence, cognitive effects, and relapse.”
  • Quality Assessment of Included Reviews: The tool used to assess review quality (e.g., AMSTAR-2, ROBIS) and key outcomes of this assessment should be reported.

>> RESPONSE: As this work was a scoping review-of-reviews rather than a systematic review or meta-analysis of primary studies, our objective was to provide a high-level synthesis of the current evidence base across both invasive and non-invasive neuromodulation modalities, rather than to formally rate the methodological rigor of each included review using tools such as RoB-2, ROBIS, or AMSTAR. However, we have tried to address this concern in two ways. First, we have updated the final paragraphs of Results subsection Sections 3.1 to more clearly highlight methodological limitations reported within the included reviews—such as small sample sizes, heterogeneity in protocols, short follow-up periods, and publication bias. Second, we have expanded the Limitations section of the Discussion to acknowledge the absence of a formal risk of bias assessment and to justify our approach in the context of the study’s aims and scope.

Discussion

  • Highlight consistent patterns, contradictions, or gaps across reviews.

>> RESPONSE: We revised the Discussion to highlight key patterns across reviews, including more consistent effects on craving than cognitive outcomes, and modality-specific findings (e.g., rTMS vs. tDCS). We also noted contradictions in cognitive domains (e.g., risk-taking vs. executive function) and gaps related to sample heterogeneity, outcome measures, and follow-up duration.

  • Contextualize findings by comparing them to recent primary studies not captured in the included reviews, other umbrella reviews on related topics, current clinical guidelines.

>> RESPONSE: We changed the discussion significantly to incorporate commentary comparing our results to prior umbrella reviews and meta-analyses in addiction and depression. We also discussed clinical translation in light of the FDA approval of rTMS for smoking cessation and implementation barriers, which are not yet addressed in current SUD treatment guidelines.

  • Highlight the strength of the evidence

>> RESPONSE: Throughout the revised Discussion, we describe which findings were most robust—particularly craving-related outcomes in stimulant, opioid, and tobacco use disorders. We also differentiate between higher-confidence results (e.g., larger meta-analyses with consistent findings) and more preliminary areas (e.g., DBS case series, cognitive effects).

  • Translate findings into practice recommendations

>> RESPONSE: We included discussion of how stimulation parameters (e.g., frequency, site, current intensity) relate to clinical outcomes, and noted the importance of optimizing these parameters for specific substances and symptoms. We also referenced the potential utility of neuromodulation as a complementary intervention when more traditional interventions have not yielded the desired outcomes.

  • Identify future research directions

>> RESPONSE: A dedicated limitations and future directions section was added, calling for standardization of protocols, inclusion of more diverse and comorbid populations, and multisite trials with longer follow-up. We also noted the need to explore how psychosocial context may interact with neuromodulation to influence craving and treatment engagement.

Tables

  • Table 1 & 2: Authors are recommended to ensure that all abbreviations used in tables are defined in the table legends. This aids reader comprehension without needing to cross-reference the text.

>> RESPONSE: Thank you for noting this, it has now been corrected for both tables.

Figures

  • Figure 2: Add a legend for this figure

>> RESPONSE: we apologize for this omission, it has been added.

Round 2

Reviewer 2 Report

Comments and Suggestions for Authors

Dear Editor,

The authors addressed all comments adequately. Thus, the manuscript could be published in its revised and improved form. 

Best regards